# Beliefs and Perceptions in Attending the Cervical Screening: The COMUNISS Project Experience

**DOI:** 10.3390/cancers17020190

**Published:** 2025-01-09

**Authors:** Narcisa Muresu, Illari Sechi, Mariangela Valentina Puci, Marco Dettori, Andrea Piana

**Affiliations:** 1Medical Management, Hygiene, Epidemiology and Hospital Infection, University Hospital of Sassari, 07100 Sassari, Italy; madettori@uniss.it (M.D.); piana@uniss.it (A.P.); 2Department of Medicine, Surgery and Pharmacy, University of Sassari, 07100 Sassari, Italy; illasechi@uniss.it; 3Clinical Epidemiology and Medical Statistics Unit, Department of Medicine, Surgery and Pharmacy, University of Sassari, 07100 Sassari, Italy; mvpuci@uniss.it

**Keywords:** human papillomavirus, cervical cancer, health education, health belief model

## Abstract

Educational interventions in health can empower individuals to make informed decisions about cervical cancer screening. Misunderstandings about personal risk and misinterpretation of screening results can decrease participation rates. To successfully implement tools like vaginal self-sampling on a large scale, it is crucial to provide tailored education and training to both the target population and healthcare professionals, ensuring better adherence to preventive programs and addressing common barriers effectively.

## 1. Introduction

Cervical cancer is the fourth most common cancer in women globally, with over 660,000 new diagnoses and 348,000 deaths in 2022, mainly caused by high-risk genotypes of human papillomavirus (Hr-HPV) [1]. Even though the greatest impact of HPV-associated diseases is related to cervical cancer, an increasing trend is registered for head-neck and anogenital cancers [2,3]. Primary and secondary prevention strategies, including HPV vaccination and cervical screening, have been implemented to reduce the number of infections and cervical lesions. Despite the effectiveness of these measures, participation in prevention campaigns remains below the targets set by the “90–70–90” goals of the World Health Organization, which expect the achievement of 90% vaccination coverage among adolescents, 70% screening uptake, and 90% treatment of cancerous and precancerous lesions [4].

Recently, the positive trend observed during the first decades after the preventive strategy’s introduction has slowed down, probably due to changes in the sexual habits of the younger generation and to sociodemographic factors, resulting in an increased circulation of the virus [5]. Moreover, the COVID-19 pandemic has almost totally paralyzed ongoing primary and secondary prevention programs and worsened existing inequities in cervical cancer screening among racial and ethnic minorities, non-heterosexual groups, rural populations, immigrants, and those with lower socioeconomic levels. Addressing these gaps in underserved populations is crucial to enhancing early detection and reducing the incidence of advanced cervical cancer cases [6].

The European Union, in response to WHO’s proposals regarding the cervical cancer elimination strategy, has developed Europe’s Beating Cancer Plan, which suggests several specific actions aimed at preventing all malignancies, including cervical cancer. These actions include improvements in the areas of prevention, diagnosis, treatment, and quality of life of patients undergoing treatment or survivors. Moreover, key points are the implementation of access to primary and secondary prevention strategies, quality and uniformity of screening programs, ascertaining the diagnostic accuracy of the available diagnostic methods, and offering free and easy access to vaccination [7].

Along with organizational barriers, individual factors lead to low levels of knowledge and awareness about the risks of HPV infection and preventive tools, which are key points in limiting access and willingness to undergo screening and vaccination. Numerous studies have demonstrated that educational interventions can avoid removing cultural and social barriers and, consequently, increase the screening participation rate [8,9].

In particular, this point has been emphasized when referring to new approaches in the prevention of cervical cancer, such as that offered by the use of vaginal self-collection as a sampling tool for HPV-DNA tests. This method has amply demonstrated its accuracy compared with clinician-collected samples, but its use remains limited if not supported by appropriate education and training targeted to both women and healthcare personnel [10,11,12,13].

Likely related to the COVID-19 post-pandemic delays, participation in cervical screening has recorded a declining trend nationally and internationally, highlighting the need to investigate factors involved in screening uptake, with a particular focus on perceived susceptibility, awareness, benefits, and barriers in participation in screening programs for targeted women [14]. This scenario supports the adoption of new solutions to increase screening uptake, such as the possibility of implementing the use of vaginal self-collection in screening programs.

Here, we evaluated the women’s beliefs and knowledge about cervical cancer and preventive measures aimed at identifying determinants in screening uptake for the first time in our setting. Our results could be useful for planning future enlarged studies that support adequate communication strategies and tailored awareness campaigns.

## 2. Materials and Methods

### 2.1. Study Design and Setting

A cross-sectional survey was conducted in the North Sardinian (Italy) female population targeted by the screening program through a web-based format. The designated geographic area accounts for over 470,000 inhabitants, of whom 120,000 were female-targeted for cervical screening (i.e., women between 25 and 65 years of age), of whom approximately 36% regularly participate in the organized screening program. To date, the first level of screening consists of the completely free offer of first-level screening and subsequent HPV-DNA for undetermined abnormalities, as well as the management of cases. In case of negativity, women are recalled after 5 years. The study setting includes small towns and villages with limited healthcare access, making it an ideal area for evaluating interventions like HPV self-sampling or targeted health education campaigns [15].

The “COMUNISS” project included an initial phase for the creation of a dedicated website (www.comuniss.it) with a “questions and answers” session providing information on the incidence, prevention, diagnosis, and treatment of cervical lesions and a session dedicated to filling out the online questionnaires.

The present study did not require ethical approval for its observational design according to Italian law (Gazzetta Ufficiale no. 76 dated 31 March 2008).

### 2.2. Data Collection

The anonymous questionnaires were administered between April and November 2023, after validation provided by a sample group of twenty experts in preventive medicine and HPV-related cancers who completed the questionnaire and offered positive feedback on its clarity, relevance, and usability. Moreover, a small group of women tested the questionnaires to verify their comprehensibility. Finally, the consistency of the responses was evaluated by Cronbach’s Alpha coefficient, obtaining an acceptable value (Cronbach’s Alpha = 0.7).

The direct link to the website and questionnaires was forwarded via social and instantaneous messaging services and by dissemination of printed content with the corresponding QR code, using a snowball non-probability sampling method, whereby initial participants were asked to share the survey with individuals within their personal and professional networks, facilitating wider reach and participation, particularly among the target demographic.

Before survey completion, each participant was asked to fill out the informed consent and agreement to collect and analyze data anonymously. According to the objectives of the survey, the questionnaire was divided into three areas of investigation: six questions about sociodemographic information and an item related to previous participation in cervical screening to assess the effect of individual variables in screening adherence; fifteen items for the HBM test, which investigated perceived susceptibility, severity, benefits, barriers, cues to action, and self-efficacy related to cervical cancer prevention; and a nine-question test, in true/false format, to ascertain the state of knowledge regarding the prevention and diagnosis of HPV-related cancers, with a focus on vaginal self-collection as a method for HPV-DNA testing (Appendix A). Answers to each question were not mandatory.

Following completion of the questionnaire, women had the opportunity to request a vaginal swab at their home, to be returned to the reference laboratory for HPV-DNA testing by self-collection. Each kit included a self-sampling swab, instructions for use, a format to be filled in with personal data (name, surname, date of birth), and an email for sending the result, plus a stamped envelope for returning to the laboratory.

### 2.3. The Health Belief Model’s Test

The Health Belief Model (HBM) is a psychological framework used to understand and predict health behaviors by focusing on individual attitudes and beliefs [16]. This model is widely applied in designing health promotion and disease prevention programs, encouraging individuals to adopt healthier behaviors through six key constructs:Perceived Susceptibility: Belief about the likelihood of getting a disease or condition.Perceived Severity: Belief about the seriousness of the condition and its potential consequences.Perceived Benefits: Belief that taking a specific action will reduce susceptibility or severity.Perceived Barriers: Belief about obstacles to performing the recommended action (e.g., cost, convenience).Cues to Action: Triggers or prompts that motivate behavior change, such as symptoms or public health messages.Self-Efficacy: Confidence in one’s ability to take the necessary action successfully.

Each item was assigned a value from the Rickert scale ranging from “disagree”, “neutral”, and “agree”, with a corresponding score of 1 to 3, respectively. An average value was calculated for each question and comprehensively for the six constructs.

### 2.4. Statistical Analysis

Collected variables were summarized using mean and standard deviation (SD) or median and interquartile range (IQR) and by absolute and relative (percentages) frequencies. Differences between women who regularly participate in screening programs (i.e., every 3 years as expected by institutional recommendation) and women who do not regularly participate/never participate in screening were evaluated by using Pearson or Fisher exact tests (qualitative variables) and the Mann–Whitney test (health beliefs total score). Univariate analysis was assessed to explore the association between Health Beliefs Model scores and adherence to the screening program: data were reported as the odds ratio (OR) and 95% confidence interval (CI). A *p*-value of less than 0.05 was considered statistically significant; data analysis was performed with STATA version 17 software.

## 3. Results

### 3.1. Sociodemographic Characteristics

A total of 1048 new users were registered between April and November 2023 for a total of 7732 accesses. The response rate for the web-based questionnaires among website users was 24.4% (256/1048). The survey questionnaires were collected from northern Sardinia with the participation of 90 municipalities out of 92 belonging to the designated area.

Women who participated in the survey, including sociodemographics and HBM items and test of knowledge, had a median (IQR) age of 36 (29–45). Below 8% of women had a lower level of education, whereas over 44% had a degree or higher education qualification. A total of 25% (63/252) of interviewers were students, and 61% (154/252) were employed. Most women had a partner (56%, 140/249) (Appendix A).

Among the 252 respondents, 130 (51.6%) regularly underwent screening for cervical cancer, whereas 74 women (29.4%) did not perform the screening test regularly, and 19% (48/252) declared they had never screened.

We found statistically significant differences in adherence to screening programs by age group, education level, and occupational status (*p*-value < 0.001), with the highest participation in employed women and those with high levels of education, whereas the lowest was registered in women under 30 years old. Differently, civil status and level of knowledge did not influence the screening uptake (Figure 1) (Appendix A).

### 3.2. Results of the Health Belief Model Questionnaire

The Health Belief Model test comprehensively evidenced a high perceived severity regarding cervical cancer condition with a median (IQR) score for all questions of 2.5 (2.25–2.75). Similarly, women showed a relevant perception of the benefits of screening tools (median (IQR) score: 3 (3–3)) and concerning the intention to take action to improve one’s health status (median (IQR) score: 3 (3–3)). Results regarding the perceived susceptibility highlight a moderate/high perception among women, with the majority of women in disagreement with the construct’s statements (median (IQR) score of 1.67 (1.33–1.67)). Individual factors (e.g., fear, anxiety, embarrassment, or pain while performing the test) do not appear to be potential barriers to screening participation (median (IQR) score: 1.5 (1–2)) (Appendix A).

We did not find any statistically significant differences in the results of the Health Belief Model’s test by age group. However, it has been evidenced that there is a lower perception of screening benefits in younger women, whereas older interviewers declared a high perception of individual barriers.

The univariate analysis carried out to assess the relationship between HBM score for each construct and adherence to the screening program revealed that a higher score in perceived barriers and severity of diseases is associated with a decrease in screening participation (OR (95% CI): 0.59 (0.39–0.87, *p*-value = 0.01), OR (95% CI): 0.39 (0.17–0.89, *p*-value = 0.03), respectively). Differently, a higher score in perceived benefits and cues to action is associated with an increased chance of undergoing screening tests (OR (95% CI): 2.41 (1.16–5.24), *p*-value = 0.02; OR (95% CI): 3.40 (1.30–8.89), *p*-value = 0.01, respectively) (Table 1).

### 3.3. Test of Knowledge and Willingness to Submit to HPV-DNA Test by Self-Collection

Overall, the women interviewed answered correctly concerning the prevention, diagnosis, and treatment of HPV-related lesions. However, it appears that the only ambiguous results were concerned with the role of HPV in extra-cervical lesions, prevention in the male gender, and knowledge about vaginal self-collection as a screening method (Appendix A).

Over 70% of women (175/245) required a self-collection kit at their home to carry out the HPV-DNA test. Forty-three (24.6%) women sent the kit back to the laboratory for analysis. The results of the HPV-DNA test found positivity to at least one HPV genotype in a quarter of the tested samples (11/43, 25.6%) with a prevalence of HPV-16, HPV-31, and HPV-68, followed by HPV-52 and HPV-51.

## 4. Discussion

Our study investigated factors involved in predicting screening behavior in women involved in cervical screening programs to address factors that could be considered in the implementation of educational strategies and in the identification of gaps in knowledge.

The decline in adherence to screening programs is partly due to post-pandemic delays and, additionally, to changes in habits and attitudes among the younger generation, which needs to be investigated, as well as the adoption of prompt actions [17]. Moreover, the emergence of new screening methods that would greatly increase coverage in the target population, such as the vaginal HPV-DNA self-test, necessitates adequate information to prepare women and optimize its effectiveness. The “COMUNISS: Know to Prevent” project is based on the evidence that planned educational interventions are key points to increasing cervical screening uptake, addressing knowledge gaps, and enhancing awareness of the importance of the early detection of precancerous lesions. Consequently, collecting this information is imperative for planning appropriate communication strategies and awareness campaigns.

The Health Belief Model is a framework used to investigate factors involved in health behaviors, and it has been used in health promotion and disease prevention since 1950 [18]. HBM has been widely used in different medical areas and has been widely applied in the analysis of determinants of adherence to cervical screening, considering the likelihood of acquiring diseases (perceived susceptibility), the understanding of the condition’s seriousness (perceived severity), the belief in the advantages of preventive actions (perceived benefits), and the recognition of obstacles to action (perceived barriers). Furthermore, the constructs include the triggers prompting behavior change (cues to action) and confidence in achieving outcomes (self-efficacy) [19].

Our results showed that sociodemographic factors and perceptions regarding individual susceptibility, emotional or organizational barriers, and a lack of knowledge about prevention strategies are determinants in the decision to participate in prevention campaigns. Particularly, high educational level and employed status appear positively involved in screening uptake. Data in the literature show that these factors play a significant role in predicting the risk of cervical cancer, as well as other diseases. The socioeconomic level, indirectly linked to the level of education and occupational status, is known as one of the most reliable predictors of health conditions [20,21]. Drolet and colleagues have highlighted that foreign women, albeit in more advantaged environments, have a higher risk of developing precancerous and cancerous lesions than the local population [20]. Moreover, as evidenced by a recent Canadian survey, household income and educational level influence the screening uptake, as well as the distance to screening centers [21]. This latter feature, such as accessibility to information and healthcare centers, becomes essential to reduce social inequalities and ensure access to prevention tools for the entire female population.

Considering the structural and emotional barriers related to the discussion of sexually transmitted diseases, it has been observed that multiple health education approaches have proved effective in encouraging screening uptake and promoting preventive health behaviors. Strategies such as reminder calls and mailed postcards are direct approaches, while mother-daughter educational sessions and small group discussions leverage family and friends’ influence to promote health across generations. Multimedia tools, including videos, presentations, brochures, and web-based platforms, facilitate reach and convenience, especially among younger people [22].

The results of the HBM test have suggested relevant questions in women’s perceptions about the prevention and treatment of cervical cancer, as well as the belief that HPV infection is directly associated with the occurrence of cervical cancer. Comprehensively, the women interviewed demonstrated a satisfactory perception of susceptibility regarding the disease, a high perception of the seriousness of the disease and its implications, a high perception of the benefits of secondary prevention strategies, and a moderate perception of the emotional barriers involved in the decision to attend the screening. However, some questions have raised gaps in understanding and confidence in the natural history of the disease and the effectiveness of treatments. Two-thirds of women wrongly considered the positivity of HPV testing as a serious problem for their health status, and nearly half of women state they are uncertain (i.e., “neutral”) about the effectiveness of treatments for cervical lesions. Issues regarding test interpretation, particularly in programs that are moving toward conversion to HPV DNA testing as the first level in screening, are essential to prevent the onset of unnecessary concerns in women who receive a positive result [21]. Previous experiences in Australia and the UK have shown significant gaps in understanding the rationale behind the transition from cytology to HPV-based screening, such as increased screening intervals (i.e., for 3 to 5 years) and interpreting HPV test results, which required clear communication by education campaigns and healthcare training [23,24]. At the same time, tailor-made pathways that support women through the course of examinations and treatment can inspire confidence in patients and influence the outcome of therapy.

Over 44% of women perceived a low risk of having acquired HPV infection in the past, particularly in older age groups. A decreased perception in this group could easily translate into a reduced screening uptake and a higher risk of developing cervical lesions, considering that the majority of infections take 15–20 years for the carcinogenesis process [25].

Regarding perceived barriers, approximately 20% of women associate a state of fear and anxiety related to the gynecological visit and the performance of the screening test. Noteworthy, an equal number of women are “neutral” on the same topic, implying a possible level of uncertainty or doubt about the feelings associated with screening participation. Personal barriers (e.g., embarrassment, fear, anxiety), along with weaknesses in screening organization, represent the main factors involved in poor adherence [13,21].

By comparing HBM’s findings with screening participation, it emerges that a high perception of the advantages of prevention and a higher propensity to act are associated with greater participation. Conversely, the high perception of emotional barriers is more frequent in the under-screened population. Remarkably, although the majority of women had a high perception of the seriousness of diseases and clinical consequences, it has been observed that there is a low correlation to screening participation. Previous studies evidenced that there are no differences between screened and under-screened women in perceived severity and confidence in clinical treatment [26,27]. Moreover, the same study suggests that more efforts should be focused on highlighting the benefits of screening and addressing cultural beliefs. Similarly, enhancing socioeconomic factors, such as employment opportunities, education levels, household income, and living conditions, could further improve screening uptake [26]. In-depth studies should investigate the underlying factors that prevent at-risk groups from participating in cervical cancer screening programs.

The application of new diagnostic and/or prognostic tools in cancer prevention is now increasingly finding its way into basic research [28]. The use of self-collected samples for high-risk HPV testing is a reliable tool with a diagnostic accuracy comparable to that of the test performed by the clinician. Moreover, self-collection provides greater accessibility by enabling screening in non-clinical settings, making it especially valuable for reaching underserved populations, women in remote areas, or those who do not regularly participate in organized screening programs [11]. Literature data have provided successful validation of the preclinical phase, transport, storage, and analysis of self-collected specimens but still leave shortcomings regarding the best strategy for the invitation of women, restitution of specimens, and management of follow-up in HPV-positive women [13,29]. The main strategies used for the implementation of self-collection are the “opt-in” (i.e., the invitation for women to order the kit) and “opt-out” alternatives (i.e., sending the kit directly to the home), with contrasting results [30]. More than 70 percent of the interviewed women requested the home kit for vaginal self-testing, even though many were not aware of this tool for performing HPV testing. However, the return rate was around 25%. This finding was not surprising compared with reports in the literature, where the return rate ranged between 8% and 20%, underlying the need to improve the delivery and offer methodologies to ensure the most optimal compliance [31]. The meta-analysis of Daponte and colleagues highlights that face-to-face invitations for self-sampling kits achieved the highest participation rates, likely due to boosting women’s confidence in performing the test, and that the “mail-to-all” approach was more successful in increasing uptake than the “opt-in” model, especially in high-income countries. Contextually, there is clear evidence of the effectiveness of vaginal self-sampling in increasing screening coverage in hard-to-reach women [32]. However, the opportunity to choose between self- and standard sampling could determine concerns and uncertainties in the target female population, who instead need a recommendation from their medical doctor or gynecologist [33]. Some studies have evidenced that offering self-sampling kits within healthcare services can significantly improve screening rates [34]. Collaboration with local agencies and community organizations, not only healthcare personnel, plays a crucial role in effectively reaching and retaining participants and improving the percentage of those who complete the screening process.

The large-scale implementation of self-collection in the screening program needs to take into consideration different delivery systems for the kit, the accuracy of the test, and which groups of women to address.

A notable limitation of the present study lies in the reliance on self-reported data, which can introduce recall or response bias, where answers may be influenced by social desirability or misunderstanding of the questions. These factors can potentially affect the accuracy and reliability of the findings, making it challenging to draw definitive conclusions and limiting the generalizability and applicability of the results to broader populations. Although the study involved individuals in the whole survey area, with 90 municipalities represented, the single-center design restricts its inference, further confined to individuals with internet access and usage. Moreover, the sampling method has intrinsic limitations due to the potential risk of selection bias and low heterogeneity in the sample study, which can lead to a non-representative sample. This raises concerns about the representativeness of the sample, particularly for populations with limited digital access or differing socioeconomic and cultural contexts. These factors should be considered when interpreting the findings and assessing their broader applicability.

Regardless of the limitations of the study, our results confirm the need to implement the knowledge level in the female population regarding the natural history of HPV infection, the risks associated with other HPV-related cancers in both sexes, and the effectiveness of primary and secondary prevention actions. New instruments such as self-collection will provide important support for cervical cancer prevention when accompanied by tailored communications and a stakeholder network that can educate women about the benefits and safety of self-collection and ensure accessibility in underserved populations.

## 5. Conclusions

These preliminary findings underscore the critical need to enhance women’s understanding of HPV’s natural history, including its potential to cause various cancers in both sexes and the value of prevention measures. Integrating self-collection as a tool for cervical cancer prevention can significantly boost screening participation, particularly when paired with well-structured educational efforts. Tailored communication strategies and robust stakeholder networks are essential to educating women about self-collection’s safety and benefits, ensuring accessibility for underserved groups, and reinforcing comprehensive prevention programs.

## Figures and Tables

**Figure 1 cancers-17-00190-f001:**
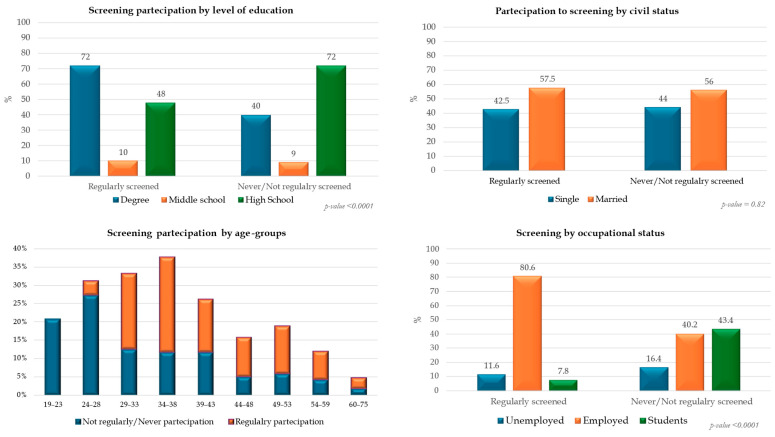
Participation in screening programs by age, civil status, level of education, and occupational status.

**Table 1 cancers-17-00190-t001:** Univariate analysis to assess the relationship between item results of the Health Belief Model test and participation in the screening program.

Variables	OR (95% CI)	*p*-Value
Perceived Susceptibility	1.49 (0.76–2.91)	0.24
Perceived Severity	0.39 (0.17–0.89)	0.03
Perceived Barriers	2.47 (1.16–5.24)	0.02
Perceived Barriers	0.59 (0.39–0.87)	0.01
Cues to Action	3.40 (1.30–8.89)	0.01
Self-efficacy	0.77 (0.50–1.20)	0.25

## Data Availability

The datasets used and/or analyzed during the current study are available from the corresponding author upon reasonable request.

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
