# Peer review of "Beliefs and Perceptions in Attending the Cervical Screening: The COMUNISS Project Experience"

_cancers, 2025, doi:10.3390/cancers17020190_

Round 1

Reviewer 1 Report

Comments and Suggestions for Authors
  1. Reliance on Self-Reported Data: The study's reliance on self-reported data may introduce biases, including recall bias or social desirability bias, which could affect the accuracy of the findings. Objective measures or more robust data collection methods would strengthen the results.
  2. Limited Generalizability: The study’s single-center design and reliance on participants with internet access limit the generalizability of the findings. Populations without internet access or from diverse socio-economic backgrounds may be underrepresented, potentially skewing the results.
  3. Low Response Rates for Self-Collection: Despite the perceived benefits of self-testing, the study found a relatively low return rate for self-collected samples (around 25%). This suggests that while self-collection may increase accessibility, challenges remain in ensuring widespread participation.

Lack of Consensus on Optimal Self-Collection Strategies: The study acknowledges that there is no clear consensus on the most effective strategies for encouraging women to use self-collection kits. Further research is needed to identify the best methods for invitation, sample collection, and follow-up for HPV-positive individuals 

Given the technical and biomedical context of your work, I would like to recommend that you consider citing the following studies that are highly relevant to the methods and technologies discussed in your paper. These studies explore similar technological advancements and their applications in cancer diagnosis and metastasis studies, which may further enrich the understanding and framework of your research:

  1. Moharamipour, S., Aminifar, M., Foroughi-Gilvaee, M. R., Faranoush, P., Mahdavi, R., Abadijoo, H., ... & Abdolahad, M. (2023). Hydroelectric actuator for 3-dimensional analysis of electrophoretic and dielectrophoretic behavior of cancer cells; suitable in diagnosis and invasion studies. Biomaterials Advances, 151, 213476.
  2. Mehran, M., Sanaee, Z., Abdolahad, M., Mohajerzadeh, S. (2011). Controllable silicon nano-grass formation using a hydrogenation-assisted deep reactive ion etching. Materials Science in Semiconductor Processing, 14(3-4), 199-206.
  3. Nikshoar, M. S., Khayamian, M. A., Ansaryan, S., Sanati, H., Gharooni, M., et al. (2017). Metas-Chip precisely identifies presence of micrometastasis in live biopsy samples by label-free approach. Nature Communications, 8(1), 2175.

Author Response

R1:

  • Reliance on Self-Reported Data: The study's reliance on self-reported data may introduce biases, including recall bias or social desirability bias, which could affect the accuracy of the findings. Objective measures or more robust data collection methods would strengthen the results.
  • Limited Generalizability: The study’s single-center design and reliance on participants with internet access limit the generalizability of the findings. Populations without internet access or from diverse socio-economic backgrounds may be underrepresented, potentially skewing the results.
  • Low Response Rates for Self-Collection: Despite the perceived benefits of self-testing, the study found a relatively low return rate for self-collected samples (around 25%). This suggests that while self-collection may increase accessibility, challenges remain in ensuring widespread participation.
  • Lack of Consensus on Optimal Self-Collection Strategies: The study acknowledges that there is no clear consensus on the most effective strategies for encouraging women to use self-collection kits. Further research is needed to identify the best methods for invitation, sample collection, and follow-up for HPV-positive individuals

AA: We thank the Reviewer for contributing to improving our work's quality. All the issues raised by the Reviewer have been listed in the dedicated section of the manuscript. We are aware of the limitations of self-reported questionnaires and limited generalizability. However, interviews with stakeholders remain the primary and most insightful method to understand perceptions about health choices effectively and to plan tailored health education interventions. Particularly the HBM helps explain why individuals adopt or avoid certain health behaviors by offering a theoretical framework based on personal perceptions, facilitates the recognition and mitigation of perceived challenges to prevention or treatment, and enables the design of targeted educational initiatives aligned with individuals' perception. Anyway, the session on study limitations was expanded based on these suggestions. The low rate of self-prevention adherence to screening underscores the need, already reported in other studies, to implement education and communication strategies on the effectiveness and safety of self-prevention in cervical screening.

R1: Given the technical and biomedical context of your work, I would like to recommend that you consider citing the following studies that are highly relevant to the methods and technologies discussed in your paper. These studies explore similar technological advancements and their applications in cancer diagnosis and metastasis studies, which may further enrich the understanding and framework of your research: 

  1. Moharamipour, S., Aminifar, M., Foroughi-Gilvaee, M. R., Faranoush, P., Mahdavi, R., Abadijoo, H., ... & Abdolahad, M. (2023). Hydroelectric actuator for 3-dimensional analysis of electrophoretic and dielectrophoretic behavior of cancer cells; suitable in diagnosis and invasion studies. Biomaterials Advances, 151, 213476.‏
  2. Mehran, M., Sanaee, Z., Abdolahad, M., Mohajerzadeh, S. (2011). Controllable silicon nano-grass formation using a hydrogenation-assisted deep reactive ion etching. Materials Science in Semiconductor Processing, 14(3-4), 199-206.
  3. Nikshoar, M. S., Khayamian, M. A., Ansaryan, S., Sanati, H., Gharooni, M., et al. (2017). Metas-Chip precisely identifies presence of micrometastasis in live biopsy samples by label-free approach. Nature Communications, 8(1), 2175.

AA: We thank the Reviewer for this suggestion. We carefully read the suggested references, and we added the relevant contents to the discussion section.

Reviewer 2 Report

Comments and Suggestions for Authors

Well written manuscript.

The methodology used to link up women via a woman who suffered from cancer appears to be rather odd. Kindly explain as what the advantages of using this approach.

It is equally important to realise that HPV does not have gender differences. Indeed, it is now reported that commonest cause of oropharyngeal cancer is HPV in the United States of America.

It is also not clear  as to what is the 'screenable population' for HPV in the province or geographical area you have selected.

It is equally important to know if the woman has to pay for the HPV test or whether is is tested free under your health system. 

Likewise, is the management of those who test positive also free in the province?

Comments on the Quality of English Language

There are several quaint expressions: by civil status, I understand you mean marital status.

The manuscript  needs to be proof-read for untangling several quaint expressions, that may not make sense to a reader in English.

Author Response

Reviewer #2

R2: Well written manuscript.

AA: We thank the Reviewer for the time spent improving the quality of our manuscript and for the important suggestions. A point-by-point response was provided.

R2: The methodology used to link up women via a woman who suffered from cancer appears to be rather odd. Kindly explain as what the advantages of using this approach.

AA: We thank the Reviewer for having raised this point. The methodology implemented in the study did not require linking up women with a woman who suffered from cancer. This is because the study targets the screened female population, and not women with a diagnosis of cervical cancer. However, we better explained the sampling method in the method section. In particular, we underlined that we opted for a non-probability sampling technique, where the process began with a small sample group of acquaintances who referred to others by social and instantaneous messaging services (see lines 109-114).

R2: It is equally important to realise that HPV does not have gender differences. Indeed, it is now reported that commonest cause of oropharyngeal cancer is HPV in the United States of America.

AA: We thank the Reviewer for this important comment. In our work, we focused mainly on cervical cancer as being the only one for which there is screening available to date. A remarkable point of the survey is that still today there is scant information about the impact of other HPV-related diseases in different anatomical sites and both sexes. Based on this suggestion, we implemented the introduction focusing on this important issue (see lines 33-35).

R2: It is also not clear as to what is the 'screenable population' for HPV in the province or geographical area you have selected.

AA: We thank the Reviewer for this suggestion. Based on the recommendations of the regional health system, the female population targeted for screening is between 25 and 65 years of age, about 120,000 women of whom, according to recent reports, only 36% regularly participate in organized screening (methods lines 87-95).

R2: It is equally important to know if the woman has to pay for the HPV test or whether is is tested free under your health system. Likewise, is the management of those who test positive also free in the province?

AA: We thank the Reviewer for this clarification request. Screening for cervical cancer is offered free of charge to the entire target population, as well as subsequent follow-ups and case management. As per the Reviewer’s request, we added the information in the text.

Reviewer 3 Report

Comments and Suggestions for Authors

Thank you for the opportunity to review this paper. The paper discusses an interesting topic. It is generally well structured, with minor English changes that need to be made. The authors should consider the following:

1. Justification is insufficient. Why this study is necessary, why now, and what new does it have to offer to scholarship?

2. The way the results are presented is poor. More detailed tables, especially with regard to correlations between variables, should be included. 

3. Discussion was not done properly. There is limited connection with existing literature, while it is not clear what the study has contributed to scholarship. Also, at times Discussion reads like summarising results, but this is not the purpose of this important section. In Discussion authors should explain their findings in relation to the literature, compare and contrast with other studies, and highlight what is new in the study that deserves attention.  

4. The authors talk about anonymous questionnaires. However, they should provide more information about what these questionnaires tried to measure. Also, how were the questionnaires validated? Receiving feedback by experts is only one of the steps for validation? What about the rest?

5. Sampling is not described in the methodology, and it is not discussed as a limitation in the Discussion. 

Author Response

Reviewer #3

R3: Thank you for the opportunity to review this paper. The paper discusses an interesting topic. It is generally well structured, with minor English changes that need to be made. The authors should consider the following.

AA: We thank the Reviewer for this important contribution and suggestions. A point-by-point replies were provided, and relative changes were added to the manuscript.

R3 : Justification is insufficient. Why this study is necessary, why now, and what new does it have to offer to scholarship?

AA: We thank the Reviewer for this relevant comment. Organized screening programs had never achieved optimal coverage in our territory and underwent a further slowdown in the post-pandemic period. Multiple reasons concur: organizational and continuity aspects in offering the service, as well as lack of awareness and willingness to participate. Another aspect is related to the morphology of the territory; indeed, screening centers are hardly accessible to those who live in small towns far from major centers. That setting suggested the possibility of identifying personal and organizational barriers and the development of new solutions, such as self-collection. The preliminary results highlighted the strengths and limitations of this approach, guiding future actions. A dedicated sentence was reported in the introduction section (Lines 73-79).

R3: The way the results are presented is poor. More detailed tables, especially with regard to correlations between variables, should be included. 

AA: We thank the Reviewer for this comment. The manuscript is presented as a “Communication”, and we followed the editorial instructions. In particular, Communications are “short articles that present groundbreaking preliminary results or significant findings that are part of a larger study over multiple years…”. According to these indications, Table X was added to Supplementary Materials (S1).

R3: Discussion was not done properly. There is limited connection with existing literature, while it is not clear what the study has contributed to scholarship. Also, at times Discussion reads like summarising results, but this is not the purpose of this important section. In Discussion authors should explain their findings in relation to the literature, compare and contrast with other studies, and highlight what is new in the study that deserves attention.  

AA: We thank the Reviewer for this important suggestion. We implemented the discussion following the more challenging contents displayed by our preliminary results. Accordingly, the comparison with the available literature was expanded, paying attention to the editorial requirements for Communications.

R3:  The authors talk about anonymous questionnaires. However, they should provide more information about what these questionnaires tried to measure. Also, how were the questionnaires validated? Receiving feedback by experts is only one of the steps for validation? What about the rest?

AA: We thank the Reviewer for this request for clarification. Based on the aims of the study, we collected information regarding demographic characteristics and existing adherence to screening programs. Moreover, the HBM tool was used to assess the women's perception and identify potential areas of intervention to improve knowledge and awareness regarding HPV-related diseases.

As we mentioned, the main tool of the investigation is the HBM, whose construct is standard. The additional parts of the questionnaire concern demographics and knowledge. In addition to the stated expert validation, pre-testing was done on a sample of women, and Cronbach's alpha (value of 0.7) was calculated to determine the reliability of the questionnaire.

The information was not included in the text, and the data are unpublished. However, following the reviewer's request we have included these additional details on the validation process in the methods (see lines 105-108).

R3: Sampling is not described in the methodology, and it is not discussed as a limitation in the Discussion. 

AA: We thank the Reviewer for this comment. We opted for a non-probability sampling technique, where the process began with a small sample group of acquaintances who referred to others through social and instantaneous messaging services. Despite the declared sampling selection bias, this method allowed us to reach a hard-to-access population, representative of the study area, involving 90 out of 92 municipalities in North Sardinia. Based on this suggestion, a detailed sentence was added to the limitation of the study.

Round 2

Reviewer 3 Report

Comments and Suggestions for Authors

Thank you for considering my feedback.